# Humidity Sensors with Shielding Electrode Under Interdigitated Electrode

**DOI:** 10.3390/s19030659

**Published:** 2019-02-06

**Authors:** Hong Liu, Qi Wang, Wenjie Sheng, Xubo Wang, Kaidi Zhang, Lin Du, Jia Zhou

**Affiliations:** ASIC and System State Key Lab, Department of Microelectronics, Fudan University, Shanghai 200433, China; 16210720074@fudan.edu.cn (H.L.); 18212020034@fudan.edu.cn (Q.W.); wsheng13@fudan.edu.cn (W.S.); xbwang16@fudan.edu.cn (X.W.); 15110720079@fudan.edu.cn (K.Z.); 17112020015@fudan.edu.cn (L.D.)

**Keywords:** humidity sensor, capacitive, PI, SIDE, IDE

## Abstract

Recently, humidity sensors have been investigated extensively due to their broad applications in chip fabrication, health care, agriculture, amongst others. We propose a capacitive humidity sensor with a shielding electrode under the interdigitated electrode (SIDE) based on polyimide (PI). Thanks to the shielding electrode, this humidity sensor combines the high sensitivity of parallel plate capacitive sensors and the fast response of interdigitated electrode capacitive sensors. We use COMSOL Multiphysics to design and optimize the SIDE structure. The experimental data show very good agreement with the simulation. The sensitivity of the SIDE sensor is 0.0063% ± 0.0002% RH. Its response/recovery time is 20 s/22 s. The maximum capacitance drift under different relative humidity is 1.28% RH.

## 1. Introduction

In addition to daily applications, such as air conditioners and humidifiers, humidity sensors are widely used in industrial process control, medical science, food production, agriculture, and meteorological monitoring [1,2,3,4,5,6,7,8,9]. In industry, the many manufacturing processes, such as semiconductor manufacturing and chemical gas purification, rely on precisely controlled humidity levels. In medical science, environmental humidity needs to be controlled during operations and pharmaceutical processing. In agriculture, humidity sensors are used for greenhouse air conditioning, plantation protection (dew prevention), soil moisture monitoring, and grain storage. Furthermore, in meteorological monitoring, weather bureaus and marine monitoring applications rely on accurate humidity sensing. For modern agriculture [10] and weather stations [11,12], accurate and fast measurement of humidity is becoming more and more important. Compared to existing infrared humidity sensors, electronic humidity sensors are cheaper, lighter, and smaller, which makes them more suitable for sensor networks to feed weather models. Nonetheless, high-precision fast-response sensors are important for many fields. For instance, fast and accurate humidity measurement are critical for eddy covariance systems [13]. Hence, electronic sensors have to become faster and more accurate.

Electronic humidity sensors can be divided into resistive and capacitive [14]. Resistive humidity sensors tend to have higher gain and are usually cheaper to manufacture than capacitive humidity sensors. However, these sensors do not respond well when operating at low relative humidity (about 10% RH) because they exhibit very poor conductivity in low relative humidity environments, making it difficult to measure the output response [15]. In contrast, capacitive humidity sensors have better linearity, accuracy, and higher thermal stability than resistive humidity sensors [16,17,18,19]. A capacitive humidity sensor responds to changes of humidity by changes of the relative dielectric constant of the sensing layer, e.g., polymer film, upon water vapor absorption. Therefore, it is possible to directly detect changes in capacitance to monitor changes in humidity. Unlike resistive humidity sensor, capacitive humidity sensors respond linearly with humidity, which simplifies the sensor readout. 

Various materials can be used as humidity sensing materials, such as electrolyte [20], ceramics [21,22], porous inorganic material [23,24,25,26], and polymers [27,28,29,30]. In particular, polymers have been used as sensing materials for capacitive humidity sensors owing to their good dielectric properties arising from their microporous structure and measurable physical property changes due to water absorption. PI is among the most commonly used moisture sensing material [31] for its good mechanical strength, electrochemical stability, and flexibility [32]. It remains stable after long time exposure to the measurement environment. Furthermore, PI is a microporous material with imide groups that strongly bond water molecules, which makes the material dielectric constant very sensitive to humidity. Therefore, we used PI in the proposed capacitive sensor. 

Capacitive humidity sensors have two basic structures: parallel plate (PP) capacitance (Figure 1a) and interdigital electrode (IDE) capacitance (Figure 1b). 

In PP sensors, the upper plate is perforated by an array of holes or parallel stripes to allow water molecules from the air to reach the sensing material underneath. Since the sensing area of the PP capacitor is sandwiched between two parallel plates, the change in the relative dielectric constant of the sensing material in the PP sensors affects the overall capacitance change. Unlike PP sensors, IDE sensors usually only affect the change in the upper capacitance of the IDEs, which makes them less sensitive than PP sensors. However, the exposed sensing area of the PP sensors is smaller than for IDE sensors, which causes a slower response than for IDEs.

The IDEs are fabricated on an inert solid or flexible substrate as parallel comb electrodes that overlap each other [6,33]. IDE sensors are easier to fabricate than PP ones. The sensitive area of the IDEs is typically a few square millimeters, and the electrode gap is a few microns. The sensitivity of this type of sensor increases with decreasing pitch [34]. The electric field strength above the IDEs decreases exponentially away from the electrode surface, and becomes one-thirtieth, or even lower, of the surface value [35] after a few microns. Therefore, in the case where the gap between the IDEs is several microns, a sensing layer only a few microns thick is enough. Thanks to this layer being completely exposed to the measurement environment, the IDE sensors are faster. However, in the IDEs, only half of the electric field lines pass through the sensing layer, and the other half of the electric field lines pass through the underlying substrate. Therefore, the IDE sensors will have only half or less sensitivity (depending on the relative dielectric constant of the substrate) compared to an equivalent PP sensor [36].

It is clear that there are advantages and disadvantages of these two types of sensors. There has been a significant effort to improve the sensor structures. For example, Zhao et al. used RIE (Reactive Ion Etching) and ICP (Inductively Couple Plasma) to etch sensing materials between parallel plates of the sensors to obtain a larger contact area with the tested environment to reduce response time from 35 s to 25 s [37], but this was still slower compared to typical equivalent IDEs.

Inspired by combining the advantages of PP and IDE structures, this paper proposes a novel IDE humidity sensor with a shielding electrode under the IDEs, namely, SIDE. On the SIDE, the capacitance of the lower half of the IDEs is shielded by an additional electrode underneath the IDEs, which effectively raises the relative capacitance change as it becomes exposed to moisture. Thus, a SIDE humidity sensor combines the high sensitivity of PP sensors and the fast response (20 s) as the IDE ones.

In this work, we first verified the feasibility of the SIDE structure in the simulation software. Secondly, the thickness of the sensing layer with different electrode gaps and the dielectric thickness between the shielding electrode and the IDEs were optimized regarding the sensitivity and response speed. The SIDE sensor with optimized parameters was fabricated. The sensitivity, response time, recovery time, and stability of the sensor were measured.

## 2. Simulation of SIDE

COMSOL Multiphysics® (Stockholm, Sweden) is applied to simulate the SIDE and IDE structure. Figure 2a shows the SIDE structure. The size of this sensor is 13 mm × 6 mm with a sensing area of 1.6 mm × 1 mm. The sensor consists of a 100 nm-thick shielding electrode, a 1 µm-thick silicon dioxide dielectric layer, a standard 100 nm IDE layer, and a PI film as the sensing layer. The finger length of the interdigitated electrode is 1 mm, with the width and the gap both being 5 µm. A total of 80 pairs of IDEs are used. A 5 µm-thick PI layer is utilized as the humidity sensing layer. Since the PI’s relative dielectric constant increases linearly with humidity [38], we simulate variations of humidity by directly changing the relative dielectric constant of the PI. An IDE model with the same structural parameters as the SIDE one is implemented with the only difference being the absence of the shielding electrode.

Figure 2b shows the simulation results of the capacitance change rate (ΔC/C_0_) of SIDE and IDE under different relative dielectric constant of PI representing the humidity conditions. C_0_ is the total capacitance when the relative dielectric constant of the sensing layer is 2.9. ΔC is the capacitance difference between any other relative dielectric constant of PI and 2.9. It can be seen that under the same conditions, ΔC/C_0_ of the SIDE structure, is about 4 times bigger than that of the IDE structure, which implies that the SIDE will have much higher sensitivity than IDE with the same parameters.

The effect of the thickness of the sensing film on ΔC_max_/C_0_ is also simulated by COMSOL Multiphysics® (Stockholm, Sweden). We define that ΔC_max_/C_0_ equals to ΔC/C_0_ with the relative dielectric constant of PI at 2.9 (C_0_) and 3.7 (C_max_), which indicates the sensitivity of the sensor.

Figure 3 shows that ΔC_max_/C_0_ increases as the thickness of the sensing film increases, but flattens at higher thickness. To optimize the sensing film thickness, two facts should be taken into account. On the one hand, it is clear that when the sensing film thickness is equal to the gap between the IDEs (as those dashed lines in Figure 3), ΔC_max_/C_0_ almost reaches saturated values. There is no significant increase of ΔC_max_/C_0_ with thicker sensing film than the gap. On the other hand, the thickness of the sensing film also affects the speed of water molecules diffusing into the sensing film completely, which defines the sensor response and recovery time. Therefore, we select the optimized sensing film thickness as equal to the gap of the IDEs. Considering the laboratory conditions, we set the width and gap of the IDEs to 5 µm.

The effect of the spacing between the shielding electrode and the IDEs, i.e., the thickness of the silicon dioxide under the IDEs on the sensitivity in the SIDE structure is also studied. 

Figure 4 shows that with the increasing thickness of the silicon dioxide layer, the ΔC_max_/C_0_ increases first and then decreases, with an optimal value of the SiO_2_ thickness of 1 µm. 

There are several parameters of the optimized SIDE structure through the simulation: the gap of IDEs and spin-coated sensing film thickness are both 5 µm, and the thickness of the silicon dioxide layer is 1 µm. These parameters are used in the fabrication of the sensor.

## 3. Materials and Methods

The sensor is fabricated on a 3-inch silicon wafer according to the following steps: (a) A 2.5 µm-thick negative photoresist is patterned. (b) An e-beam-evaporated Ti/Au layer is deposited and selectively removed by a lift-off process to form the bottom shielding electrode. (c) A layer of 1 µm silicon dioxide is deposited by PECVD (Plasma Enhanced Chemical Vapor Deposition). (d) IDEs are fabricated on the silicon dioxide by the same sequence of lithography, e-beam evaporation, and lift-off. (e) A 5 µm-thick PI is spin-coated. Subsequently, the device is baked at 120 °C for 1 h, 180 °C for 1 h, and 250 °C for 6 h to cure the sensing layer. The completed sensor and cross-section of the SIDE structure under scanning electron microscope (SEM) are shown in Figure 5. The same IDE structure fabricated on the glass substrate without the shielding electrode is studied as the control experiment.

The setup for the humidity measurement is shown in Figure 6. The test is always carried out in an incubator. We build the simple incubator with heaters and semiconductor coolers inside. Each of them is controlled by an external PID (proportional integral derivative) controller to keep the temperature constant. In the incubator, we place a bottle of saturated salt solution and the sensor. The humidity is also monitored by a commercial humidity meter (Rotronic, HC2-S) at the same time and in the same incubator. The uncertainty of HC2-S is ±0.8% RH. The capacitance measurement uses an IC chip (SMARTEC’s UTI03) and additional circuits. The commercial humidity sensor and the capacitance measurement circuit communicate with the computer using serial port simultaneously. The humidity and capacitance are recorded in parallel by the computer for later analysis.

The capacitance above the shielding electrode *C*_x_ can be directly measured using the circuit shown in Figure 7 without mixing the capacitance between the shielding electrode and IDEs C_pn_ (n = 1, 2). C_x_ is the sensing capacitance proportional to the humidity. C_p1_ and C_p2_ are the capacitances between the shielding electrode and the IDEs. C_f_ is the fixed capacitance of the IC chip. U_1_ and U_2_ are the potentials before the humidity sensor and after the IC chip that both can be measured. Therefore, C_x_ can be calculated using Equation (1).
C_x_ = −U_1_/U_2_·C_f_(1)

Before the test, each device is placed in an oven at 100 °C for 10 min to get rid of the effect of the previous measurement.

The sensitivity (S) can be expressed as Equation (2):S = (ΔC/C_0_)/Δ(% RH)(2)
where ΔC = C_1_ − C_0_, C_0_ is the capacitance measured at the RH, which is 23.7% ± 0.8%, and C_1_ is the capacitance measured when the RH is 73.0% ± 0.8%. Δ(% RH) is the difference between the relative humidity values when measuring C_1_ and C_0_.

The response and recovery dynamics are among the most important characteristics for evaluating the performance of humidity sensors. The response time for RH increase and the recovery time for RH decrease are usually defined for a sensor as the time taken to reach 90% of its total capacitance variation. The response and recovery curves are measured by exposing the SIDE sensor to alternate levels of humidity between 2.0% ± 0.8% and 77.0% ± 0.8% RH.

In order to evaluate the functioning of the humidity sensor over long periods of time, we measured the sensor’s capacitance over the duration of 20 h at 25 °C with relative humidity levels of 25.7% ± 0.8%, 34.4% ± 0.8%, 45.0% ± 0.8%, 57.0% ± 0.8%, and 73.5% ± 0.8% RH.

## 4. Results and Discussion

A sensitivity test is carried out on the SIDE and IDE structure. Figure 8 shows the capacitance measured from SIDE and IDE at different levels of humidity, and their linear fits with *R*^2^ of 0.996 and 0.991, respectively. The slopes of the line, i.e., S of SIDE and IDE are 0.0063 and 0.001,65, respectively. Taking the uncertainty of HC2-S into consideration, the S of SIDE and IDE are 0.0063 ± 0.0002 and 0.001,65 ± 0.000,05, respectively. Hence, the sensitivity of the SIDE structure is 3.82 times bigger than that of the IDE. These results show the significant improvement of sensitivity brought by the shielding electrode, that minimizes the large constant capacitance of the substrate. Indeed, whatever substrate the IDE is built on, the relative dielectric constant of the substrate is larger (e.g., Si is 11.9, glass is 10) or close to (e.g., flexible polymer films) the relative dielectric constant of PI (2.9–3.7). The experimental result and simulation data verify the effects of the shielding electrode and shows high agreement as well. It is clear that our proposed SIDE structure can provide an effective way to measure relative humidity more sensitively and accurately. Another advantage of the shielding electrode is that it can effectively suppress the external electromagnetic interference and reduce the noise in the measurement process.

Figure 9 shows the responses of the SIDE sensor. The absorption curve represents the response of the sensor as a function of time, from an environment with low relative humidity to an environment with high relative humidity. The desorption curve represents the response of the sensor as a function of time, from an environment with high relative humidity to an environment with low relative humidity. The curve can switch to steady states rapidly after the RH level changes. Our sensor’s response/recovery time is 20 s/22 s, which is comparable to 1 s/15 s for normal IDE reported in the literature [39], but a little worse. This is because in their work, the thickness of the sensing film is only 0.65 µm, while ours is 5 µm. If we scale down our sensors to reduce the IDE gap, the required sensing film thickness will also decrease, resulting in great improvement in response speed. Limited to laboratory conditions, we fabricated the sensor with 5 µm gap. However, our sensor’s response/recovery time is still much better than 122 s for PP sensors [40].

Figure 10 shows the stability characteristic of the SIDE sensor. The sensor is kept in the incubator for 20 h at 25.7% ± 0.8%, 34.4% ± 0.8%, 45.0% ± 0.8%, 57.0% ± 0.8%, and 73.5% ± 0.8% RH, respectively. The magnitude of the drift of sensor capacitance is converted into the apparent changes in relative humidity, D, which is calculated by
D = (C_max_ − C_mean_)/(C_0_⋅S)(3)
where C_max_ is the maximum measured capacitance after the sensor is exposed to different RH atmosphere, and C_mean_ is the average capacitance of all recorded values at a certain relative humidity, C_0_ is the capacitance measured when the RH is 23.7% ± 0.8%. The maximum drift value (D) obtained from Figure 10 under different relative humidity was 1.28% RH. Thus, our sensor is able to achieve satisfactory stability from a practical standpoint, which makes it promising as a commercially available sensor.

## 5. Conclusions

In summary, we propose a novel shielded interdigitated electrode structure for humidity sensing. We perform a comprehensive simulation of this structure to optimize the parameters for the sensor fabrication. In simulation and actual testing, we find that the sensitivity of the SIDE structure is much higher than that of the IDE structure because of the effect of the shielding electrode on the capacitance change rate. Since the surface structure of the SIDE structure is still the same as IDE, the SIDE sensor combines the high sensitivity of the parallel plate sensors and fast response of the IDE sensors. The sensitivity of SIDE is 0.0063% ± 0.0002% RH, and the response/recovery time is 20 s/22 s. The stability of the SIDE sensor was also characterized. The maximum drift value under different relative humidity is 1.28% RH. 

Meanwhile, since the basic operating principle of many capacitive sensors is the same, the SIDE structure can even be applied to capacitive gas sensors, such as volatile organic compound (VOC) sensors which are used to monitor toxic gases. This shows that SIDE can replace IDE in various sensors that are more sensitive to the accuracy and response speed.

## Figures and Tables

**Figure 1 sensors-19-00659-f001:**
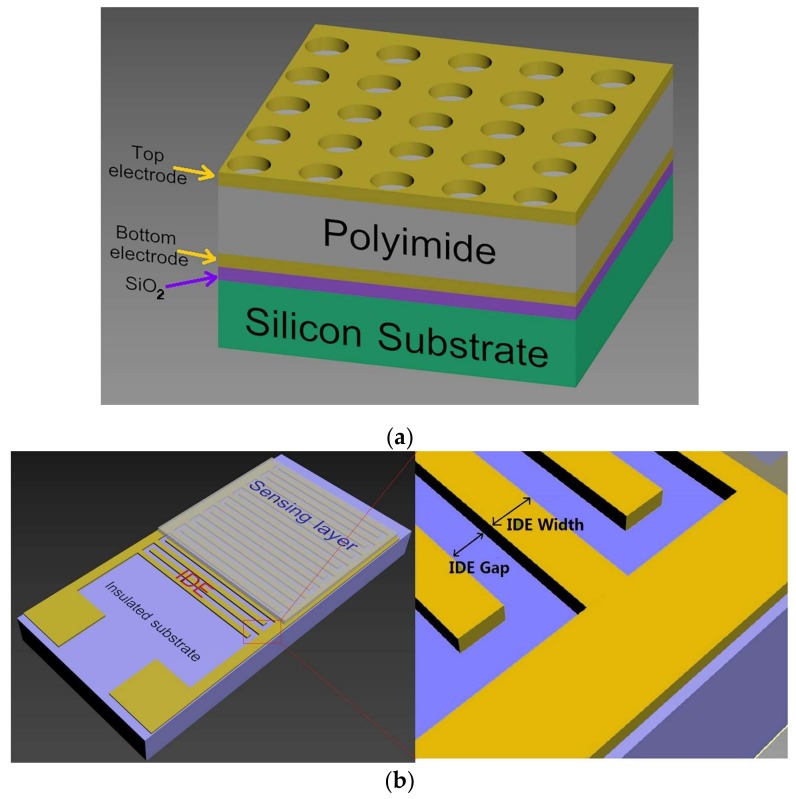
Structure diagram of parallel plate (PP) and interdigital electrode (IDE) sensors. (**a**) PP sensors composed of a solid substrate, two layers of parallel plate electrode, and a sensing material between them. (**b**) IDE sensors composed of an inert substrate, IDEs, and sensing material layer atop of the IDEs. A partial enlarged detail of IDE is shown on the right.

**Figure 2 sensors-19-00659-f002:**
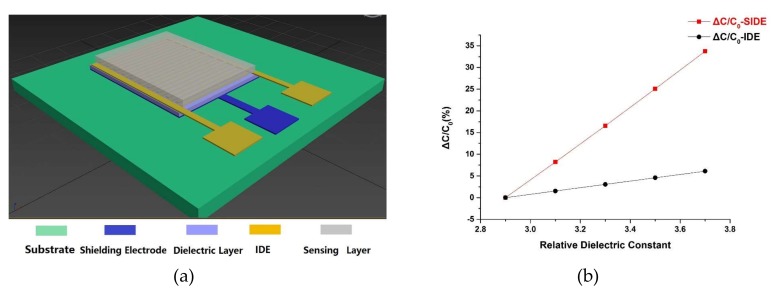
SIDE structure and simulation results. (**a**) 3D model of SIDE structure; (**b**) Comparison of the relative changes in capacitance of the SIDE (**red line**) and IDE (**black line**) structure according to numerical simulations.

**Figure 3 sensors-19-00659-f003:**
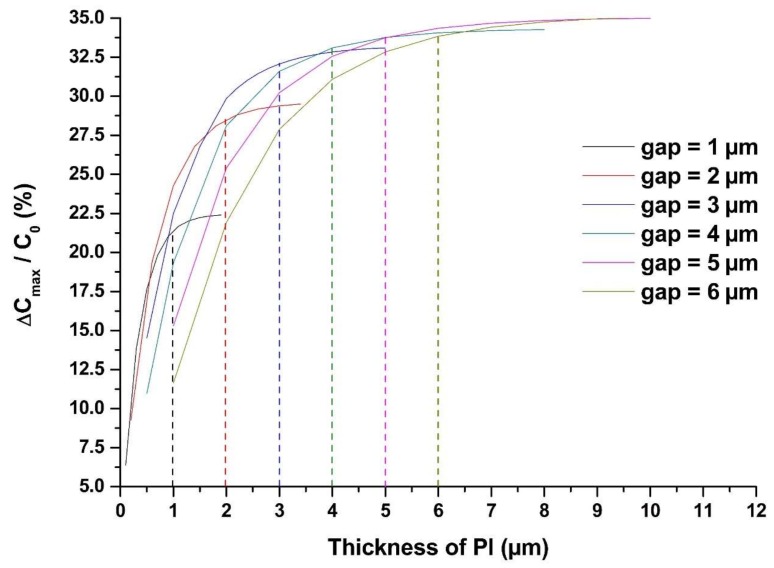
Influence of sensing film’s thickness on sensor sensitivity. The vertical ordinate of the intersection of all the dashed lines and the solid curves represents the sensor’s ΔC_max_/C_0_ when the sensing film thickness is equal to the gap between the IDEs.

**Figure 4 sensors-19-00659-f004:**
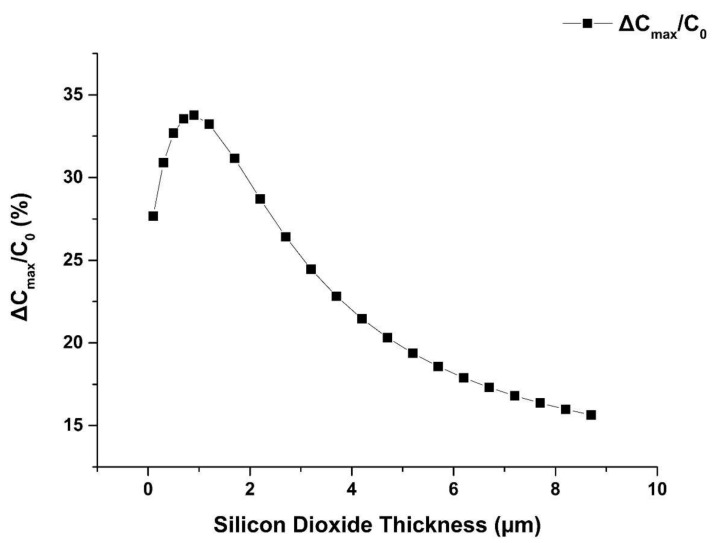
Influence of silicon dioxide thickness on the sensor sensitivity. For increasing silicon dioxide layer thickness, the full sensitivity increases first and then decreases past an optimal value.

**Figure 5 sensors-19-00659-f005:**
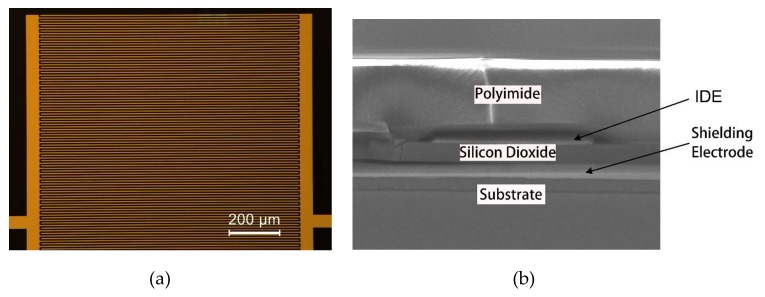
SIDE sensor picture under microscopy, and its cross-section image under SEM.

**Figure 6 sensors-19-00659-f006:**
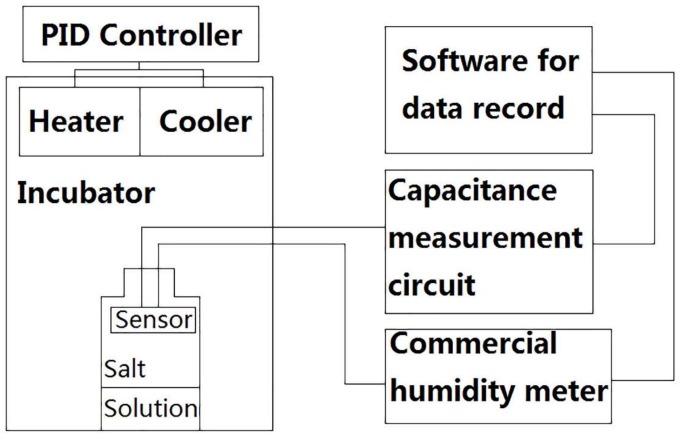
Block diagram of the measurement system consisting of an incubator, a measurement circuit and recording software.

**Figure 7 sensors-19-00659-f007:**
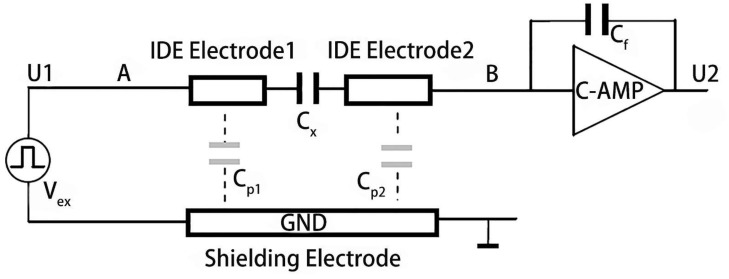
The working principle of the humidity capacitance measurement. The key point is to calculate the capacitance of C_x_ by measuring the induced charge generated at point B.

**Figure 8 sensors-19-00659-f008:**
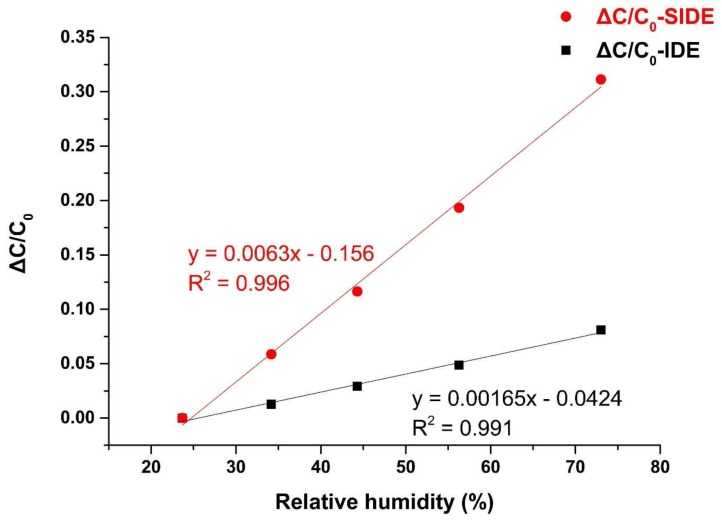
Experimental measurement of sensitivity of SIDE and IDE humidity sensors.

**Figure 9 sensors-19-00659-f009:**
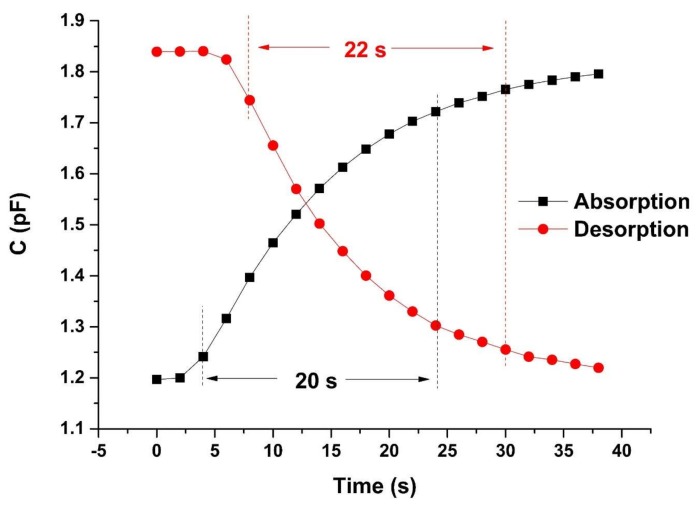
The response and recovery curves are measured by switching the SIDE sensor, alternately, between 2.0% ± 0.8% and 77.0% ± 0.8% RH. The response/recovery time is 20 s/22 s.

**Figure 10 sensors-19-00659-f010:**
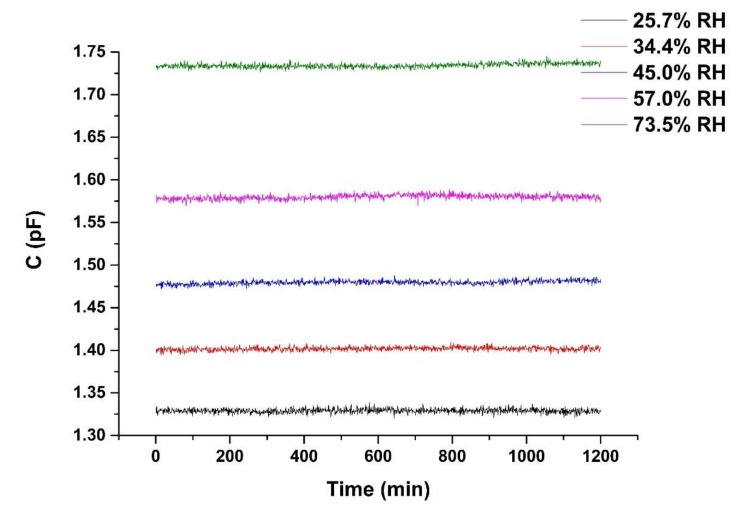
Stability of SIDE sensor. The sensor is kept in the incubator for 1200 min at 25.7% ± 0.8%, 34.4% ± 0.8%, 45.0% ± 0.8%, 57.0% ± 0.8%, and 73.5% ± 0.8% RH, respectively.

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
