# Peer review of "Humidity Sensors with Shielding Electrode Under Interdigitated Electrode"

_sensors, 2019, doi:10.3390/s19030659_

Round 1
Reviewer 1 Report
The author’s reports a study about humidity sensors with shielding electrode under interdigitated electrode.
This is a good paper. It contains very interesting information about the sensitive layer with various electrode gaps and dielectric thickness between the shield electrodes.
The following comments perhaps could help the authors to improve the manuscript:
1. In 1. Introduction section of the paper should be mentioned the information about various kind of sensors, from literature, which promising for practical use: (eg: 10.1016/j.spmi.2014.11.022; 10.1007/s11664-014-3190-6; 10.1007/s10854-015-3750-4; http://dx.doi.org/10.1016/j.apsusc.2014.02.138).
2. In 4. Results and Discussion section please compare the results from this manuscript with the reported literature for electrical behavior of similar polymers (eg: 10.1007/s13726-011-0011-0).
3. The manuscript is consisting, English is good.
Author Response
Thanks for the reviewers’ comments concerning our manuscript entitled “Humidity sensors with shielding electrode under interdigitated electrode”. Those comments are all valuable and very helpful for revising and improving our paper, as well as the important guiding significance to our researches. We have studied comments carefully and have made the correction which we hope to meet with approval. Revised portion are marked in red in the paper.

Reviewer 2 Report
Sensors-417292-peer-review-v1
The paper reported the results of development of a Humidity sensors with shielding electrode under interdigitated electrode. That was a new method to produce a new sensing element. However, serious problem need to be solved.
In abstract, the full name must be mention, such as PI, SIDE, IDE
There are two terms of humidity, relative humidity and absolute humidity. The absolute humidity is not affected by temperature, only relative humidity is. Line 32-33 “ but the humidity is influenced by other factors: atmospheric pressure, temperature and so on.” Was not correct.
Figure 6 need to be improved.
The most important problem was the standard humidity environment, in line 198-200, “In order to evaluate the functioning of the humidity sensor over long periods of time, we measured the sensors capacitance over the duration of 20h at 25℃ in the atmospheres of 25.7%, 34.4%, 45.0%, 57.0% and 73.5% RH separately.”.
How to produce and maintain these humidity environment?
How to ensure these values, 25.7%, 34.4%, 45.0%, 57.0% and 73.5%? Were they measured by other humidity sensors?
What is the uncertainty of these RH environment?
Why only use these narrow RH range? Line 39, “However, these (resistive) sensors do not respond well when operating at low relative humidity (about 10% RH)”. If this capacitance was only suit in these narrow RH range, why not select the commercial resistive sensor?
Please provide the data that RH environment was lower than 20% and higher than 90% to ensure the performance of your sensors.
Line 101-102, “The sensitivity, response time, recovery time and the stability of the sensor were measured.” If the standard RH environment could not be recognized, all report were meaningless.
Please indicated the measurement uncertainty of your sensors.
The content of introduction could be condensed. Some common sense about materials could be deleted.
Author Response

(The authors gave the same response as above.)

Reviewer 3 Report
The topic of the manuscript is important and actual. I suggest the publication after the correction of the manuscript.

Author Response

(The authors gave the same response as above.)

Round 2
Reviewer 2 Report
The content of revised version have improved. However, two questions need to be classified.
1. Please list the name of saturated salt solutions to control 5.7%, 34.4%, 45.0%, 57.0% and 73.5% that listed in the literature (HUMIDITY FIXED POINTS OF BINARY SATURATED AQUEOUS SOLUTIONS”(DOI: 10.6028/jres.081A.011)
2. The relative humidity is also verified by a commercial humidity meter (Rotronic, HC2- S). 25.7%, 34.4%, 45.0%, 57.0% and 73.5%? What is the uncertainty of these RH environment?
3. Please commend the performance of this RH sensors by considering the above uncertainty.
Author Response
We gratefully acknowledge the referees for reviewing our manuscript “Humidity sensors with shielding electrode under interdigitated electrode”. Those comments were valuable and very helpful for revising and improving our paper, as well as clarifying the significance of the work. We have reviewed the comments carefully and have made the corrections accordingly. Revised portion are marked in red in the paper. The main corrections in the paper and the response to the reviewer’s comments are in the annex.

Reviewer 3 Report
After the minor revision I suggest the manuscript for publication.

Author Response

(The authors gave the same response as above.)

Round 3
Reviewer 2 Report
The revision content have improved.